**Cite this article:** Wilson KA (2023).
Prioritisation to prevent extinction.
*Cambridge Prisms: Extinction*, **1**, e6, 1–7

biodiversity conservation; biodiversity loss;
conservation planning; Red List; triage

**Author for correspondence**:
Kerrie A. Wilson,
Email: kerrie.wilson@qut.edu.au

# Prioritisation to prevent extinction

Kerrie A. Wilson 🆔

School of Biology and Environmental Science, Queensland University of Technology, Garden's Point, Brisbane, QLD, Australia

## Abstract

Prioritisation is about choice, and in the context of species extinction, it is about choosing what investments to make to prevent extinctions as opposed to assessing extinction risk, identifying species that are doomed to extinction, or mapping components of biodiversity. Prioritised investments may focus on conservation activities aimed at species protection or management, but they may also seek to acquire new knowledge to resolve uncertainties. Two core components of prioritisation are a clearly stated objective and knowledge of what activities can be undertaken, acknowledging that there are likely to be dependencies between these activities. As the natural environment and society change, so will the enabling conditions for conservation, hence the need to be adaptable and proactive into the future.

## Impact statement

A diversity of conservation activities is needed to avert the loss of species threatened with extinction. Prioritisation of investments can enhance the transparency and defensibility of resource allocation and can also inform the funding required to reverse decline of species.

## Introduction

The Anthropocene era is dominated by a sixth mass extinction (Barnosky et al., 2011; Ceballos et al., 2015) with key drivers being global environmental change, habitat destruction and fragmentation, overexploitation, pollution, and invasive species and diseases (Prugh et al., 2010; Allek et al., 2018; Hirsch et al., 2020). The scale of the biodiversity crisis demands urgent action, with approximately 1 million species threatened with extinction globally (IPBES, 2019). A diversity of conservation activities is needed to avert the loss of species and ecosystems. While the biodiversity extinction crisis is on international and national policy agendas, the amount of funding invested to date has been insufficient to achieve global targets (McCarthy et al., 2012). Where resources require careful allocation, prioritisation can reduce the possibility of *ad hoc* or biased allocation of resources (Wilson et al., 2009). The focus of this review is on prioritisation to prevent species going extinct, and therefore the unit of interest is the species. This review has three objectives: (1) clarify what prioritisation is, and what it is not; (2) summarise the commonalities amongst approaches and points of contention; and (3) identify important areas for future debate and research.

## Core components of a prioritisation

At its most basic, prioritisation means to arrange in order of priority (Mace et al., 2007). Prioritisation is underpinned by decision science or operations research (Hemming et al., 2022), and seeks to enhance the extent to which decisions (i.e., choices between alternatives given a stated objective) are informed, transparent and defensible. In the field of biodiversity conservation, prioritisation typically informs the allocation of conservation resources (i.e., funding, effort, time and attention).

There are several approaches to species prioritisation. They differ according to whether they prioritise:

- species themselves (Chen, 2007; Liu et al., 2019),
- their habitats or populations (Nielsen and Kenchington, 2001; Clarkson et al., 2012; Strimas-Mackey and Brodie, 2018),
- conservation activities targeting species (Joseph et al., 2009; Wilson et al., 2009; Rose et al., 2016; Brazill-Boast et al., 2018; Gillespie et al., 2020),
- abatement or mitigation of particular threats more generally (Carwardine et al., 2019), or
- protection of areas of land critical for species protection (Sinclair et al., 2018; Leclerc et al., 2022).

While the focus here is on species, it is also important to consider the actions required to abate the threats to species persistence, the locations where these actions must be implemented, and the

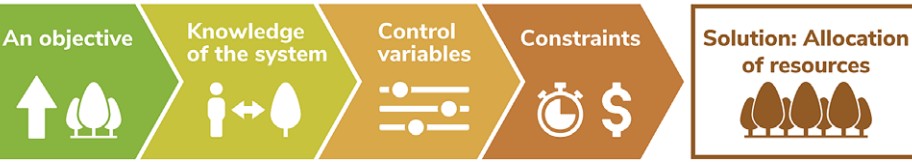

**Figure 1.** Core components of a prioritisation problem.

most effective timing for implementation (Game et al., 2013). The actions of interest will have a cost and likelihood of success, and consideration of these more pragmatic components of prioritising species has triggered debate around the use of the term "triage" in conservation (see below).

The basic conservation prioritisation problem has the following core components (Figure 1).

### An objective

This is a measurable interpretation of our overarching goal. In conservation, the objective is typically to maximise the number of species conserved (to some targeted level) or minimise the number of species that go extinct over some time period (Wilson et al., 2009). The far end of this spectrum is an objective of zero extinction (Box 1).

### Knowledge of the system

In the context of prioritising species to prevent extinctions, key knowledge requirements are the species of interest, the threats to these species, the actions that might be taken to abate these threats or improve the likelihood of the species persisting, and the cost of those actions. Importantly, species are generally threatened by multiple threats and as such there are likely to be dependencies and synergies between the threats, and between the associated mitigating actions that are available for investment (see Box 2).

### Control variables

Reflect the things that we can do, such as how much funding or effort is directed towards the actions in any location and at any

---

**Box 1.** Alliance for Zero Extinction.

Launched globally in 2005, the Alliance for Zero Extinction (AZE) focusses on sites for preventing global extinctions of species. AZE sites are often the last remaining refuges of one or more endangered or critically endangered species. The protection of such sites was aligned with the Convention on Biological Diversity's Aichi Targets, namely Aichi Targets 11 and 12, with the overarching objective of improving the status of biodiversity by safeguarding ecosystems, species and genetic diversity. The AZE sites are also aligned with the draft Post-2020 Global Biodiversity Framework, specifically (CBD, 2021), milestone A.2 (the increase in the extinction rate is halted or reversed, and the extinction risk is reduced by at least 10%, with a decrease in the proportion of species that are threatened, and the abundance and distribution of populations of species are enhanced or at least maintained). There are over 850 AZE sites worldwide, and over half of these are at least partially protected. The alliance focusses on engaging with governments, institutions, community groups, and non-governmental organisations to improve and implement policy, deliver site conservation programs and progress research efforts to prevent extinctions. From the viewpoint of the alliance, the objective of species prioritisation is zero extinction (Wiedenfeld et al., 2021).

---

**Box 2.** Threat interactions, co-extinction and other dependencies.

Species at risk of extinction are typically impacted by more than one threatening process. For example, in Australia at-risk fauna are impacted by multiple threatening processes (Allek et al., 2018). There are likely to be commonalities between required management actions to abate threats to species, and furthermore, the actions taken can interact; that is, the costs, benefits and feasibility of one action can change when another action is undertaken. Consider for example the control of invasive species. Actions taken to control an invasive species may release other pest species from competition or predation. In Australia, control of invasive rabbits alone may lead to intensified predation of native prey by foxes, whereas control of foxes alone may result in increased rabbit populations and competition with native herbivores. If these interactions are ignored, opportunities to enhance efficiency could be missed and targeted efforts could be compromised. Explicitly managing will likely alter decisions about what actions to invest in and where they should occur and has the potential to deliver increased investment efficiency (Auerbach et al., 2015; Figure 2). An extension of this approach is to assign the threats to species to particular industries or sectors in order to prioritise overarching sectoral improvements in management and ultimately accountability (Prugh et al., 2010).

---

point in time (Hughey et al., 2003). For example, we might seek to invest funds in restoring degraded habitat across multiple locations or protect an area from development.

### Constraints

These may include a budget envelope, or alternatively, the minimum amount of conservation or benefit that is sought. This minimum amount might be specified as a target. Targets can be tailored to account for life history characteristics and the habitat needs of a species, such as the minimum viable population size for a species needed for it to persist in the wild. Alternatively, targets can be generalised across species (e.g., protect 30% of the range of each species of interest).

A generalised version of the conservation prioritisation problem subject to a fixed budget, $b_t$, is described below. Let $x_{jkt}$ be the amount of money to be spent on action $k$ in location $j$ in time period $t$. Each year the cost of all the actions across all the locations must be less than our overall budget, so we have the constraint

$$\sum_{j=1}^{N}\sum_{k=1}^{P} x_{jkt} \leq b_t, \quad \text{for every year } t,$$

where $N$ is the number of locations and $P$ is the number of possible actions.

The overall aim is to find a solution (or multiple strong-performing solutions) through manipulation of the control variables that have the highest possible value of the objective function subject to our constraints. Each location $j$ has a cost $c_j$ and each asset (which might include species) $i$ has a target $r_i$. The variable $x_j$ equals

## The Objective

To maximize the number of native species that benefit from three threat mitigation actions: fire management, invasive predator (red fox; *Vulpes vulpes*) control, and reduced grazing pressure (habitat degradation) of domestic stock.

## Knowledge of the System

Seventy-two species across southeastern Queensland, Australia, were identified as vulnerable to one or more of the three threats.

## Control Variables

Managing one threat may interact with another: the cost, benefit, and feasibility of one action can change when another action is undertaken.

Different assumptions can be made about how actions interact:

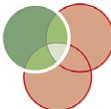

**Independent:** species benefit from an action proportional to the number of threats it faces.

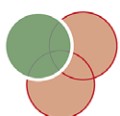

**Optimistic:** managing one threat will benefit all species, regardless of the number of threats its faces.

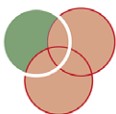

**Pessimistic:** species will only benefit from actions if all threats are managed.

Legend:
tree, shrub, cycad, sedge, climber, herb, orchid — amphibian, butterfly, fish, reptile, bird, mammal, ○ Threat

## Constraints

Land managers are acting under budget constraints and must decide which management approach to undertake.

## Solution

Expected benefits and costs (relative to a $10M investment, and an independent baseline) depends on the assumption made.

| | Optimistic | Pessimistic |
|---|---|---|
| Benefit change | +13% | -5.6% |
| Cost change ($M) | -$4.3 | +$2.6 |

**Figure 2.** Case study: Prioritisation accounting for threat management interactions. Adapted from Auerbach et al. (2015).

1 if location $j$ is selected for investment, otherwise it equals 0. The objective can be to either minimise the cost.

$$\sum_{j \in P} c_j x_j$$

subject to the constraint that the targets are met for each asset of interest.

The objective of the maximal-coverage problem is to maximise some measure of "benefit", given a fixed budget. That is, the objective is to maximise

$$\sum_{i \in I} f\left(y_i(x)\right).$$

Subject to the constraint

$$\sum_{j \in J} c_j x_j \le b.$$

A variety of algorithms can be used to prioritise investments in species (Hanson et al., 2019). These algorithms might generate optimal or near optimal solutions to a specified problem such as maximising coverage or minimising loss (Wilson et al., 2009). Alternatively, scoring approaches can be used to rank alternative options according to specific criteria, such as cost-effectiveness or cost-utility (Wilson et al., 2007). Scoring approaches will provide inefficient solutions if they do not account for the complementarity between actions (Possingham et al., 2006).

The key is to distinguish conservation problem formulation (the objective, control variables, etc.) as outlined above, the algorithms or criteria that can be used to find solutions to the problem (e.g., simulated annealing) and software packages (e.g., Marxan) that can package up all these components into an easy-to-use interface (Ball et al., 2009).

### Prioritising species versus assessing extinction risk

The process of prioritising species to prevent extinction differs from assessing extinction risk *per se*. Extinction risk is typically assessed using criteria such as species life history characteristics often within a population viability analysis framework (Liu et al., 2020),

responsibility for protection (Martin et al., 2010; Kricsfalusy and Trevisan, 2014), taxonomic uniqueness (Chen, 2007), rarity (Toledo et al., 2014), management feasibility (Martin et al., 2010), recovery potential (Di Marco et al., 2012), species distribution (Liu et al., 2019), threat status or some combination of these criteria.

The threat status of a species is often determined by the International Union for the Conservation of Nature (IUCN) Red Listing criteria, which use quantitative rules to assign the risk of extinction (Mace et al., 2008) and ecosystem collapse (Keith et al., 2013). Building on these static lists, the Red List Index evaluates trends in biodiversity (Butchart et al., 2007). Red Lists are supported within countries through legislation that seeks to protect threatened and endangered species, such as the *Environment Protection and Biodiversity Conservation Act 1999* in Australia. Red Lists have singly been used to prioritise species, but they are not a prioritisation in and of themselves (Miller et al., 2006) – they do not identify specific actions, or quantify what would be involved to shift the conservation status of a species or ecosystem (Collen et al., 2016; Kyrkjeeide et al., 2021). Red List assessments are also not comprehensive across all species or even taxonomic groups (Walsh et al., 2013; Tingley et al., 2016; Tapley et al., 2018).

The "Red to Green" framework was developed to translate the Red List Index into prioritisations based on extinction risk (Akçakaya et al., 2018; Kyrkjeeide et al., 2021). This approach uses quantitative criteria of risk assessment (i.e., extinction risk for species, risk of ecosystem collapse for habitat) to develop measurable objectives and targets for species and habitats (e.g., the improvement in the Red List category to be achieved by a target year). This is then followed by the identification of conservation actions needed to reach the goals and quantification of the costs of these actions and other constraints. This approach is much closer to a species prioritisation, compared to the use of threat status alone.

Some have approached species prioritisation by combining evolutionary data with measures of extinction risk. The focus here is to prioritise threatened species that represent large amounts of phylogenetic/functional trait diversity using metrics such as evolutionary distinctiveness (ED; Faith, 2008; Cadotte and Davies, 2010; Gumbs et al., 2018). Metrics such as ED ideally require species-level phylogenies to calculate the individual contribution of each species to the total phylogenetic diversity of a clade (Isaac et al., 2007). Such genetic data is not routinely available and species-level phylogenies are often incomplete, although imputation methods exist (Curnick et al., 2015; Gumbs et al., 2018; Weedop et al., 2019) and scenario-based approaches to account for uncertainties in extinction probability values have also been applied (Billionnet, 2017). While providing a more comprehensive assessment of the benefit of taking action to mitigate the threats to a species, this approach alone falls short of a comprehensive prioritisation if it does not have a specified objective or account for what needs to be done where and when in order to secure the persistence of the species of interest.

## Triage controversy

Prioritising species also carries with it an ethical dilemma (Wilson and Law, 2016), reflected in the controversial debate about the use of the term "triage" in conservation. Triage in the field of emergency medicine is a process of prioritisation under severely constrained resources, where the needs of a few, resource-intensive, critical cases are "sacrificed" so that resources can be distributed to a greater number of less critical cases. Controversy associated with

---

**Box 3.** Conservation prioritisation and flagship species.

Similar to the arguments against triage, there has been a long-running debate in conservation about the use of flagship species. An overreliance on such flagships (e.g., koala, tiger and panda) can be seen to divert resources from less charismatic or well-known species. However, such flagships can be important for raising funds for conservation (Veríssimo et al., 2017), and thus filling the shortfall in funding needed to prevent species from going extinct (see section "Triage controversy"). There is the possibility that carefully selected flagship species can raise funds for conservation and also encourage spending of resources to conserve broader biodiversity. Key places that harbour at least one charismatic flagship species but that also maximise a broader biodiversity objective have been identified (McGowan et al., 2020). Through such integrated conservation planning analyses, it is possible to maximise public awareness and raise funding for conservation while still achieving broader species conservation goals. Ward et al. (2020) demonstrated a comprehensive prioritisation analysis using umbrella species – that is, species that due to their large habitat requirements can facilitate the protection of other naturally co-occurring species (Roberge and Angelstam, 2004). Ward et al. (2020) demonstrate that the umbrella species approach can also be used in conjunction with a systematic prioritisation of funds to protect and recover species at risk of extinction through an analysis of the spatial distribution of threats, of conservation actions and costs, as well as overlaps in species geographic ranges.

---

the use of the term in conservation has been well summarised previously (Bottrill et al., 2009) and it is clear from the discussion that has ensued over the past decade that the fundamental concern about the use of the word "triage" in conservation relates to how the term is interpreted.

"Triage" in conservation has typically been interpreted as allowing some critically endangered species to go extinct to save others. Its enaction is assumed to reveal the species that have been relegated to extinction, with some arguing that the concept of triage should be avoided altogether in conservation so as not to preclude opportunities to expand the resources available for conservation (Wiedenfeld et al., 2021), whether it be for protection or monitoring (Wheeler et al., 2016). Alternatively, we should view triage as a process that analyses the expected outcomes of investments in different species, which is then used to prioritise investment based on what can be achieved with different levels of investment or effort. Such an analysis can then be used to determine the investment needed to prevent most species from going extinct, instigating efforts to fill any gap, such as through conservation marketing and fundraising campaigns (see Box 3). But while triage can inform, and motivate, investment in conservation, ultimately the level of resources available to prevent extinctions is a socio-political decision.

## Asset maps shows what we value, but are not necessarily priorities

The identification of priority places for species conservation has received significant attention in both the peer-reviewed and grey literature (Moilanen et al., 2009). The focus here has been on mapping the distribution of biodiversity and its patterns, such as centres of species endemism (Orme et al., 2005), uniqueness (Eken et al., 2004), biodiversity hotspots (Myers et al., 2000) and various measures of diversity (Grenyer et al., 2006; Brum et al., 2017). The production of these asset maps rests on the assumption that protecting these places from the threats to species persistence in these locations will result in the best outcomes for biodiversity conservation at large. While such maps may highlight the uneven

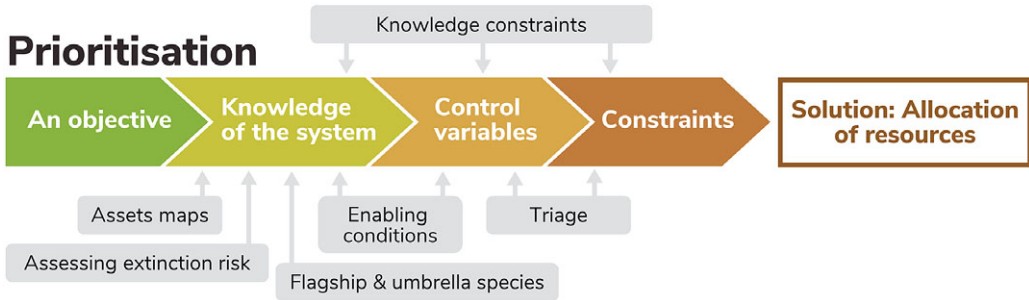

**Figure 3.** Additional components of the prioritisation problem, with a focus on key additional elements in relation to species prioritisation.

distribution of biodiversity and areas that could yield returns from investment, since the identification of these places is not situated in a properly formulated problem (with an objective, constraints, etc., as outlined above) it is not possible to discern whether these places are indeed conservation priorities, nor the relative priority of one place compared to another.

### Broader contextual considerations and challenges

Prioritising species to prevent extinctions needs to consider not only extinction risk, but a range of enabling conditions that influence the likelihood of success of conservation investments such as financial, cultural, logistical, ethical, human livelihoods and social factors (Miller et al., 2006; Fitzpatrick et al., 2007; Moir and Brennan, 2020). Furthermore, there are often conflicting objectives (Simmons et al., 2021), differing value judgements (Latombe et al., 2022), varying risk preferences and tolerances (Tulloch et al., 2015) and sources of uncertainty. Given all these complexities, it is important to utilise a variety of theories and methods from the decision science toolbox (Hemming et al., 2022).

Knowledge constraints are often posed as an argument against prioritisation: to not prioritise species reduces the risk of misallocating effort. There is indeed uneven knowledge of threatened species, with a research bias towards larger bodied species and charismatic or economically valued species (Allek et al., 2018); within taxonomic groups, there is often a bias towards species occurring in developed nations (Buechley et al., 2019). Building on the pre-cautionary principle the use of the best available knowledge and prioritising further research as well as implementation of conservation activities is key. Importantly, the very act of determining the most appropriate conservation action (or suite of actions) might be considered as a control variable if these actions are unknown at the time of prioritisation (Game et al., 2013; Raymond et al., 2018). Decision science methods based on the value-of-information theory to determine and appraise the relative value of further data collection versus managing species at risk of extinction to determine optimal strategy are also available (Bennett et al., 2018). Similarly, proactively progressing activities to prevent species from being at risk of extinction in the first place is critical (Walls, 2018).

As knowledge improves or as values and perceptions change, priorities are likely to shift. Furthermore, funded projects may underperform or new projects may require investment when additional funding is not available immediately (Gerber, 2016). Reallocation of ongoing investments in response to shifting priorities (i.e., reprioritisation) would need to be balanced by the likely transaction costs, lost opportunities and the risks associated with

not honouring the needs of ongoing project commitments (Wu et al., 2021).

### Conclusions

The biodiversity crisis is immense – around 1 million species are under threat globally – and the funding currently available to prevent extinctions is inadequate. As such, prioritisation of conservation efforts is essential. A wide range of approaches to prioritisation has been used, and this is appropriate given the diversity of contexts and the pace of environmental change, nonetheless, all prioritisations should be based on sound knowledge of the systems in question and clearly defined objectives, control variables and constraints (Figure 3). Careful prioritisations can improve decision making and motivate further investment in conservation, the funding available to prevent extinctions is a socio-political decision.

**Open peer review.** To view the open peer review materials for this article, please visit http://doi.org/10.1017/ext.2023.3.

**Data availability statement.** Data availability is not applicable to this article as no new data were created or analysed in this study.

**Acknowledgements.** I am grateful for the review of early drafts of this manuscript by Dr. Blake Simmons, Dr. Ilva Sporne and Timothy Campbell. I would like to thank Dr. Jess Hopf at Knowlegible Designs for providing the figures. This work was supported by ARC SRIEAS Grant SR200100005 Securing Antarctica's Environmental Future.

**Author contributions.** The author confirms sole responsibility for the following: conception of the study, manuscript preparation and revision.

**Financial support.** This research received no specific grant from any funding agency, commercial or not-for-profit sectors.

**Competing interest.** The author declares none.

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
