## [Reviewer Report]

*Comments to Author*: Comments on Wilson “Prioritisation to prevent extinction”

This is an interesting and potentially useful review of the of the prioritisation concept and its application to decision making in the context of species conservation. Nevertheless, I consider the ms is a bit too theoretical, relying too much on abstract concepts such as the core components of prioritisation and very little (or nothing) on practical examples. For example, I think that the definition of control variables needs to be more operative, so than a decision maker can identify them in a more straight-forward way. The same would apply to concepts such as “budget envelope” or “minimum amount of conservation or benefit”. Therefore, I consider the ms and its take-home message would greatly benefit from the reference to one or several case studies that illustrate theoretical concepts, as well as from some graphical depiction of the theoretical framework of prioritisation.

One particular case study that comes to my mind is that of the relic population of capercaillie (Tetrao urogallus) of the Cantabrian mountains in northern Spain (please, do not get me wrong, I am NOT suggesting the use of this example, because I do not even know if it would be fit; you surely know dozens of them that are much more suitable). In this area, capercaillie is a relic of glacial time, isolated from all other Eurasian populations and adapted to much milder and temperate climate conditions (in terms of diet preferences, for example), so that it is classified as a separate subspecies. Hunting, land-use change, human disturbance and climate change have pushed Cantabrian capercaillie into the vortex of extinction, but the species Tetrao urogallus is quite abundant and widely distributed across the Eurasian belt of boreal forests, from Scandinavia to Siberia. Should the Cantabrian subspecies be prioritised at world level? At Spanish level? How should it be ordered in the priority list in relation to other threatened species that are farther from extinction but show more restricted distributions (even endemic species) and worrying declining trends? Perhaps the ms would gain insight if you dived into this kind of aspects.

Apart from this, you discuss the utility of the flagstaff species concept. I suggest you expand a little more as to include the umbrella species concept, which may partly overlap with the former, and has also been widely used in conservation prioritisation.

---

## [Reviewer Report]

*Comments to Author*: As a curator of living collections with a conservation remit, rather than a conservation biologist, I found this review paper both informative and thought-provoking. I believe it will be a useful contribution in clarifying what is meant by conservation prioritization, and in setting out distinctions between various activities and decisions taken on by conservationists in the broader sense and thereby aiding their overall contribution to the positive outcomes society needs. I only have minor comments with regard clarity in a couple of sections.

The following section I found particularly difficult to parse. Rephrasing, breaking it up, or even examples might help:

“They differ according to whether they

50 prioritise species themselves (Chen, 2007; Liu et al., 2019), their habitats or populations

51 (Clarkson et al., 2012; Nielsen and Kenchington, 2001; Strimas-Mackey and Brodie, 2018),

52 conservation activities targeting species (Brazill-Boast et al., 2018; Gillespie et al., 2020;

53 Joseph et al., 2009; Rose et al., 2016; Wilson et al., 2009), abatement or mitigation of

54 particular threats more generally (Carwardine et al., 2019) or protection of areas of land

55 critical for species protection (Leclerc et al., 2021; Sinclair et al., 2018).”

The following section is rather damning of a widely known part of the literature, such as relating to the widely used phrase ‘biodiversity hotspot’. I see the point of the criticism in this context, but might it be worth expanding somewhat on the real value of such results: e.g., in highlighting the uneven distribution of biodiversity globally; indicating areas within which conservation prioritization might yield disproportionate impact?

“since the identification of these

226 places is not situated in a properly formulated problem (with an objective, constraints, etc,

227 as outline above) it is not possible to discern whether these places are indeed conservation

228 priorities, nor the relative priority of one place compared to another”

---

## [Reviewer Report]

*Comments to Author*: Thank you very much for the opportunity to review manuscript EXT-22-0003, “Prioritisation to prevent extinction”. I very much enjoyed reading this review and I think it sets the stage very well for the audience of this new journal. The review is well structured and argued. I only have one small suggestion to make below. Other than that I think the manuscript is in great shape and would make an important contribution to the journal.

Line 78 and 81. For both control variables and constraints I would recommend adding a bit of text especially to introduce some examples so that readers new to the topic get a better idea of what control variables and constraints could be.

Best regards,

Richard Schuster

Director of Spatial Planning and Innovation, Nature Conservancy of Canada

Email: richard.schuster@natureconservancy.ca

---

## [Editor Report]

*Comments to Author*: The paper is useful, and all three reviewers enjoyed reading it. However more examples are needed, and more background to more theoretical elements to really demonstrate the potential of the approaches discussed

---

## [Reviewer Report]

*Comments to Author*: I think authors have followed my suggestions and the ms is now much clearer and focused. I have no further remarks.

---

## [Reviewer Report]

*Comments to Author*: I reviewed this ms. in its first version and had only minor comments. These have been addressed satisfactorily. The additional graphics are a welcome further improvement. I look forward to seeing this published.

---

## [Reviewer Report]

*Comments to Author*: Thank you very much for the opportunity to review manuscript EXT-22-0003.R1, “Prioritisation to prevent extinction”. I very much appreciated reading this revised version of the manuscript. In my mind all reviewer comments have been addressed sufficiently and I don’t have any further comments to make at this point. I’m happy to recommend acceptance of this fine work.

Best regards,

Richard Schuster

Director of Spatial Planning and Innovation, Nature Conservancy of Canada

Email: richard.schuster@natureconservancy.ca